# Trichalcogenasupersumanenes and its concave-convex supramolecular assembly with fullerenes

Yixun Sun [1,2], Xin Wang[1,2], Bo Yang[1,2], Muhua Chen[1], Ziyi Guo[1], Yiting Wang[1], Ji Li[1], Mingyu Xu[1], Yunjie Zhang[1], Huaming Sun[1], Jingshuang Dang[1], Juan Fan[1], Jing Li [1] ✉ & Junfa Wei [1] ✉

Synthesis of buckybowls have stayed highly challenging due to the large structural strain caused by curved $\pi$ surface. In this paper, we report the synthesis and properties of two trichalcogenasupersumanenes which three chalcogen (sulfur or selenium) atoms and three methylene groups bridge at the bay regions of hexa-*peri*-hexabenzocoronene. These trichalcogenasupersumanenes are synthesized quickly in three steps using an Aldol cyclotrimerization, a Scholl oxidative cyclization, and a Stille type reaction. X-ray crystallography analysis reveals that they encompass bowl diameters of 11.06 Å and 11.35 Å and bowl depths of 2.29 Å and 2.16 Å for the trithiasupersumanene and triselenosupersumanene, respectively. Furthermore, trithiasupersumanene derivative with methyl chains can form host-guest complexes with $C_{60}$ or $C_{70}$, which are driven by concave-convex $\pi\cdots\pi$ interactions and multiple C−H$\cdots\pi$ interactions between bowl and fullerenes.

Bowl-shaped polycyclic aromatic hydrocarbons (PAHs), often referred to as buckybowls or $\pi$-bowls, have emerged as attractive targets that captivate scientists from chemistry to materials science in light of their intriguing characteristics as well as potential applications in a diverse of scientific fields[1–5]. More significantly, some of the buckybowls could also serve as templates or seeds for the growth of single-walled carbon nanotubes[6–9] having a controlled chirality and diameter and thus having uniform electronic properties useful in molecular electronic devices[10]. Owing to the huge inner strain of bowl-shaped hydrocarbons makes their synthesis a major challenge, the number of buckybowls is still relatively rare compared with flat PAHs. Hitherto the most literature-known hydrocarbon buckybowls were related to $C_{60}$[11–19] or $C_{70}$[20–26] fullerene fragments and their $\pi$-extended derivatives[27–30]. In addition, some heteroatoms-embedded buckybowls involving B[31], N[32–38], P[39], S[40], Se[41–43], etc. have also been synthesized and sought as model compounds and partial structures for higher heterofullerenes.

In the past decades, our group has been involved in the design and synthesis of new PAHs, naturally, bowl-shaped polyarenes have never escaped our consideration. It is of course interesting to imagine that if each of all six bay regions of hexa-*peri*-hexabenzocoronene (*p*-HBC, also known as superbenzene) are bridged by one divalent group, it should formally constitute a large, compact, highly symmetric, zigzag rimed, and bowl-shaped architecture composed of 48 atoms and 19 rings in its bowl system (Fig. 1). Such bowl molecules feature a coronene core successively circumscribed by alternate hexagonal and pentagonal rings, we prefer to call it (hetero)supersumanene as a convenient trivial name based on similar structural characteristics and larger size compared with sumanene[12]. Simple Chem3D simulation and normative density functional theory (DFT) calculation indicates a beautiful bowl-shaped geometry for these supersumanene derivates (i.e., X, Y = C, S, Se, etc.). However, experimentally bringing these imagined molecules into existence by chemical synthesis represents a gigantic challenge. Müllen and Feng et al. have disclosed an elegant synthesis of *tri*-sulfur annulated *p*-HBC derivatives (TSHBC)[44] in which only three five-membered rings formed in bay regions, further full S annulation were not observed due to the high strain. Fortunately, during this paper was being reviewed, Tan group reported the synthesis of *hexa*-selenium annulated *p*-HBC derivatives (HSHBC)[43] by using

[1]School of Chemistry and Chemical Engineering, Shaanxi Normal University, Xi'an 710119, China. [2]These authors contributed equally: Yixun Sun, Xin Wang, Bo Yang. ✉e-mail: li_jing@snnu.edu.cn; weijf@snnu.edu.cn

a similar strategy to TSHBC. The success of HSHBC can be attributed to the bigger atomic radius of selenium atom and smaller strain. These results fully reflect the difficulty to construct such curved supersumanene structures with smaller atoms, such as C and S. Herein, we disclose the synthesis and properties of two classes of trichalcogenasupersumanenes with three chalcogen (sulfur or selenium) atoms and three methylene groups alternately embedded at the bay regions of *p*-HBC.

## Results and discussion
### Synthesis and characterization
Figure 2 depicts our three-step synthetic route for constructing these trichalcogenasupersumanene derivatives. Our synthetic campaign was commenced with aldol cyclotrimerization of 1-(9,9-dibutyl-2,7-dichloro-9*H*-fluoren-4-yl)ethan-1-one (**2**), which was prepared from commercially available 2,7-dichloro-9*H*-fluorene in three steps (see Supplementary Information Section 1.2). The initial attempts towards the cyclotrimerization of **2** using SiCl₄[45] as a catalyst in ethanol failed; only a trace amount of the desired product, 1,3,5-tris(9,9-dibutyl-2,7-dichloro-9*H*-fluoren-4-yl)benzene (TFB, **3**), was detected in the reaction mixture by MS spectroscopy. Fortunately, several explorations rewarded us with finding proper conditions to access the cyclotrimer **3** by performing the reaction in *o*-dichlorobenzene (*o*-DCB) at 180 °C under microwave irradiation using TiCl₄ as catalyst[16]. This venerable method, albeit under slightly harsh conditions, delivered the product **3** in 52% isolated yield after chromatographical purification. Noticeably, TFB has the carbon structure that constitutes the entire carbon scaffold of final bowls and the functionalities necessary for the formation of the five-membered rings.

Also fortunately and delightfully, the consequent Scholl reaction of **3** with 1,2-dichloro-4,5-dicyanobenzoquinone (DDQ) and trifluoromethanesulfonic acid (TfOH) in 1,2-dichloroethane at 50 °C, which is the crucial step toward the total synthesis of the designed bowls, afforded the hunted trifluorenocoronene (TFC, **4**) in an isolated

yield of 37%. Although not high, the achieved yield is quite reasonable if considering the high ring strain from three five-membered rings and steric hindrance of two chlorine atoms at each bay region in the product **4**. The preinstalled *n*-butyl groups endow TFC with adequate solubility amenable to isolation and full spectroscopic characterization. All analytical data are consistent with the expected structure of TFC **4** (Supplementary Figs. 67 and 68). We also briefly explored the improvement of Scholl reaction of **3** including temperature, reaction time, and stoichiometry of acid, and finally found relatively satisfactory conditions, as denoted in Fig. 2. It is noteworthy that the presence of chlorine atoms is also critical to the success of the Scholl reaction, replacing the chlorine atoms with the hydrogen atoms will lead to complex products. Moreover, the hexfluorinated and hexabrominated versions of key intermediate TFC **4** could not be obtained via oxidative cyclization.

With the requisite TFC **4** in hand, we then accomplished the total synthesis of the designed buckybowls via threefold heteroatom annulation at its dichlorinated bay positions using Stille type reaction, which has been established by Wang group in their synthesis of sulfur heterocyclic annulated perylene bisimide derivatives[46] with some modifications. To our delight, we obtained successfully the hunted buckybowl, trithiasupersumanene **1a**, as a yellowish powder in 58% isolated yield using (Bu₃Sn)₂S as the sulfur donner and Pd(PPh₃)₄ as the catalyst at 150 °C in a sealed tube under Ar atmosphere. The HR-MS spectrum showed the molecular peak at *m/z* 985.3936 consistent with the desired product (C₆₉H₆₀S₃+H⁺, 985.3930) and also, the measured isotopic distribution was well coincident with that stimulated (Supplementary Fig. 71). The ¹H NMR spectrum (Fig. 3) presented only one sharp low field singlet at 8.06 ppm for the six equivalent aromatic protons; while the high field signals, which are well resolved, implied two inequivalent sets of *n*-butyl groups. These observations strongly suggest its 3-fold symmetry and the bowl-shaped conformation in solution since only if access was gained to the bowl structure, the aliphatic chains can be differentiated by their location at regions demarcated by the convex and concave faces of the bowl. The chemical shifts of butyl chains inside the concave follow an order of $H_a > H_c > H_d > H_b$, similar to 9,9-dibutyl-9*H*-fluorene[47]. While butyl chains outward the bowl follows a different order of $H_{a'} > H_{b'} > H_{c'} > H_{d'}$. Interestingly, two distinctive signals, $H_b$ and $H_d$, assignable to methylene and methyl protons in the butyl moieties inward bowl orientations manifest negative chemical shifts (−1.04 to −0.02 ppm), which can be ascribed to the stronger shielding effect caused by the ring current of the bowl system. The ¹³C NMR data further support the bowl topological character of **1a** (Supplementary Fig. 70).

The selenium version was also synthesized by bridging the bays with selenium atoms under the same conditions but switching (Bu₃Sn)₂S to (Bu₃Sn)₂Se[48], as a yellowish powder in 39% isolated yield. Similarly, all spectroscopic data (Supplementary Figs. 72, 73 and 74) are well consistent with the expected structure of triselenosupersumanene

**Fig. 1 | Our synthetic concept of bowls based on *p*-HBC.** Each of bay regions of *p*-HBC are bridged by one divalent group (X or Y), supersumanene are defined as X = Y = C, while heterosupersumanene are defined as X,Y = heteroatom.

**Fig. 2 | The synthetic route for the trichalcogenasupersumanenes 1a and 1b.** Reagents and conditions: (i) 6.6 equiv. TiCl₄, *o*-dichlorobenzene, (microwave), 180 °C, Ar, 3 h; (ii) 8 equiv. DDQ, 5% TfOH, 1,2-dichloroethane, 50 °C, Ar, 2 h; (iii) 4 equiv. (Bu₃Sn)₂S or (Bu₃Sn)₂Se, 1 equiv. Pd(PPh₃)₄, toluene, 150 °C, Ar, 48 h.

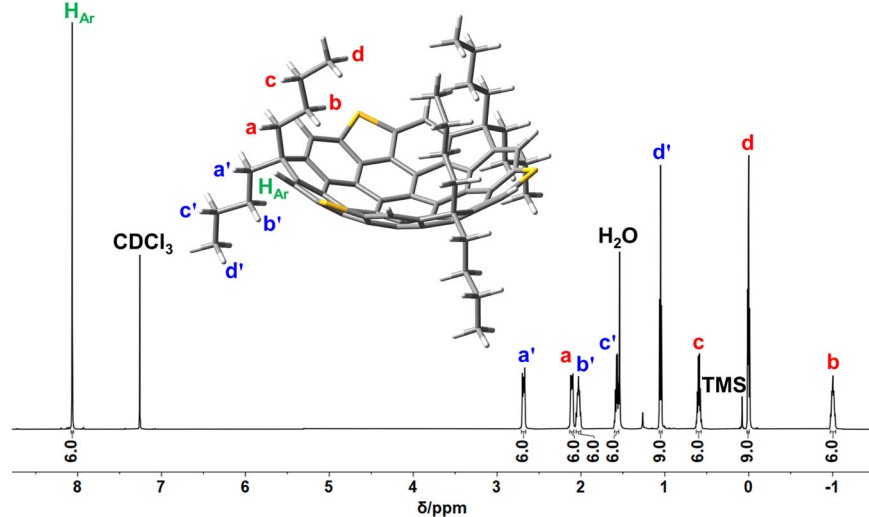

**Fig. 3 | ¹H NMR Spectroscopic characterization of 1a.** ¹H NMR Spectra (600 MHz) and labeling of protons on **1a**, CDCl₃, H₂O, and tetramethylsilane (TMS). Inset: The geometry predicted by DFT calculation and the labeling of protons on **1a** are defined.

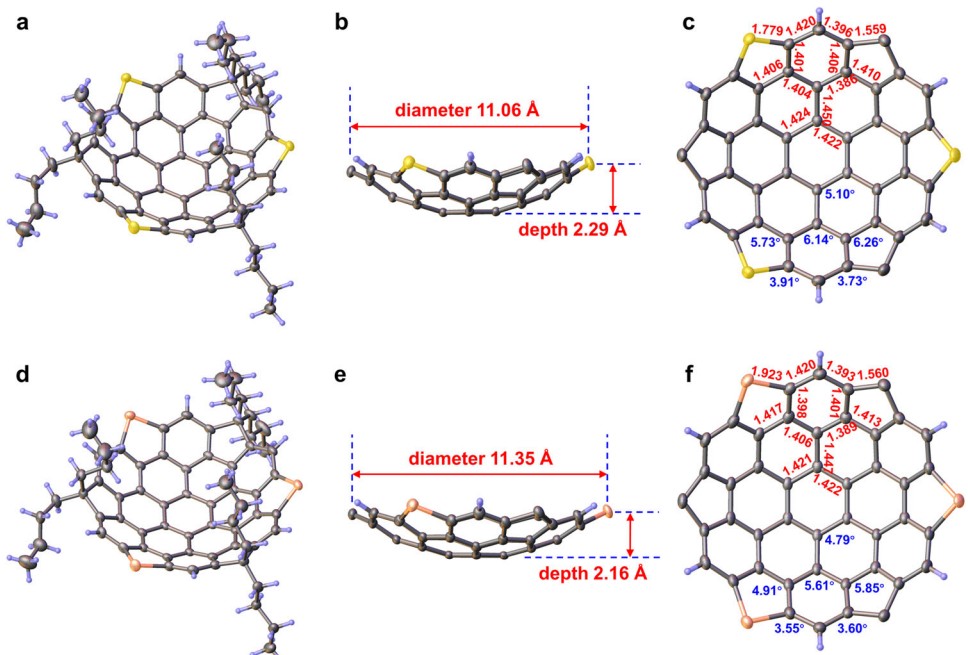

**Fig. 4 | X-ray crystal structures of 1a and 1b.** The perspective views of **1a** (**a**) and **1b** (**d**). The side views of **1a** (**b**) and **1b** (**e**) with diameters and depths (red). The top views of **1a** (**c**) and **1b** (**f**) together with POAV angles (blue) and mean bond lengths (red). Solvent molecules and butyl groups in side and top views have been omitted for the sake of clarity. Thermal ellipsoids are shown at 30% probability.

**1b.** It should be pointed out that its ¹H NMR and ¹³C NMR spectra show virtually the same pattern as the sulfur analog. Furthermore, the corresponding CH₂(b) and CH₃(d) signals at the higher field (−0.96 to 0.01 ppm) due to the weaker shielding effect, reflecting that the bowl is flatter than its sulfur analog.

Thermogravimetric analysis (TGA) experiments were conducted to determine the thermal stability of both compounds. The results showed that the onset decomposition temperatures of both **1a** and **1b** are above 400 °C (Supplementary Fig. 24), reflecting their thermal robustness. Meanwhile, these trichalcogenasupersumanenes are stable enough and can be stored in solution and solid under air atmosphere for weeks without any change.

## X-ray crystallographic analysis

Single crystals of **1a** and **1b** suitable for X-ray diffraction studies were grown by slow vapor diffusion of methanol into a solution of chloroform at room temperature. The X-ray structural analyses unambiguously confirm the anticipated bowl-shaped architecture (Fig. 4a, d), with an approximate $C_{3v}$-like symmetry in both crystals; the deviations from ideal symmetry should be blamed to the crystal packing forces in the crystalline state. The maximum bowl depths and diameters are 2.29 Å and 11.06 Å for **1a** while 2.16 Å and 11.35 Å for **1b**, defined by the perpendicular distance between the plane of three peripheral heteroatoms and the centroid of the hub and by the horizontal distance between saturated carbons and heteroatoms at opposite vertexes,

respectively (Fig. 4b, e). Deservedly, the selenium bowl is slightly shallower than its sulfur isolog due to its bigger atomic radius. It is worth mentioning that the depths and widths as well as other structural parameters of the bowls are closely aligned with those of DFT calculations (Supplementary Figs. 38 and 39). Normal concave-convex stacking fashion is restricted due to the presence of three pairs of *n*-butyl groups, while edge-to-convex predominant stacking manner stabilized with C−H ··· π and chalcogen ··· π interactions were observed (Supplementary Figs. 3 and 12). Both bowls exhibit concave to concave packing motifs along *a* axis via six C−H ··· chalcogen hydrogen bonds between chalcogen atoms and hydrogen atoms (H$_d$) of methyl groups inward bowls (Supplementary Figs. 4, 5, 13 and 14).

As shown in Fig. 4c, f, there are six categories in a total of 36 different pyramidalized trigonal carbon atoms that existed in these trichalcogenasupersumanenes, except for 6 negligible pyramidalized $sp^2$ carbon atoms at the peripheral vertexes. The π−orbital axis vector (POAV) angle analyses[49] show that the carbon atoms of the hub ring of two bowls are moderately curved with a mean POAV angle of 5.10°(**1a**) and of 4.79°(**1b**); the most curvature occurs at the carbon atoms making up the coronene rim, of which the mean POAV angles are 6.26°(**1a**) and 5.85°(**1b**) for carbon atoms fusion with cyclopentadiene rings. The carbon atoms linked to each hub ring are pyramidalized significantly with a mean POAV angle of 6.14°(**1a**) and of 5.61°(**1b**), the secondary maximum value of those of the carbons in respective molecule. The curvature is evenly distributed among the peripheral carbons attached to heteroatom and to the saturated carbon atom of cyclopentadiene ring, of which POAV angles are 3.91°(**1a**) and 3.55°(**1b**) and 3.73°(**1a**) and 3.60°(**1b**), respectively (Supplementary Figs. 9 and 18).

The bond lengths, as denoted in Fig. 4c, f (red colored data), are the symmetry-averaged values of all equivalent bonds around their respective $C_3$ symmetric axis disregard deviations from perfect geometry. The hub ring in both molecules remains an almost regular hexagon with a negligible bond length alternation (BLA) of 0.002 Å (**1a**) and 0.001 Å (**1b**) and equalized yet elongated bond lengths of 1.422 Å and 1.424 Å (**1a**) or 1.421 Å and 1.422 Å (**1b**) in comparison with that in benzene (1.40 Å), indicating their reduced aromatic bond character. The radial bonds joining the hub to the rim hexagons are elongated to 1.450 Å (**1a**) and 1.447 Å (**1b**), considerably longer than the others in the same ring and thus indicative of quasi-single bond character. These bonds are not components of aromatic rings and function only as geometric connections between the hub and outer rings. Accordingly, the six-membered rings containing the radial bonds deviate markedly from regular hexagons. Six rim benzene rings in both compounds are irregular hexagons with the bond lengths spanning from 1.386 Å to 1.420 Å (**1a**) and from 1.389 Å to 1.420 Å (**1b**), falling within the scope of aromatic bond character. The C−S bond (1.779 Å) in **1a** and C−Se bond (1.923 Å) in **1b** are fairly longer than that in dibenzothiophene (1.744 Å)[50] and in dibenzoselenophene (1.895 Å)[51], indicating also declined aromatic bond character at the corresponding positions. In addition, the fluorene moieties show a C−C$_{sat.}$ bond length of 1.559 Å (**1a**) and 1.560 Å (**1b**), longer than that in 9,9-dioctyl-9*H*-fluorene (1.523 Å)[52]. These results lend support to that the peripheral and hub benzene rings possess more benzenoid character, consistent with the deductions from BLA analysis based on the X-ray determined (Supplementary Figs. 8 and 17) and DFT calculated structural data (Supplementary Fig. 39). The peripheral and hub benzene rings of **1a** and **1b** possess the smallest BLA values, which are similar to the parent HBC (Supplementary Fig. 19). In addition, both **1a** and **1b** display elongated bond lengths for the hub and periphery but shortened bond lengths for the rim of coronene core in comparison with parent HBC (Supplementary Fig. 20) due to the high inner strain of their bowl geometries.

## Conformational analysis

The aromaticity distribution characteristics in the bowl systems were further evaluated by nucleus independent chemical shift (NICS)[53] and anisotropy of the induced current density (ACID)[54] calculations on their unsubstituted analogs **1a**' and **1b**' at B3LYP/6-311G(d,p) level of theory. The calculated NICS(1)$_{zz}$ values, as denoted in the corresponding rings (Fig. 5a, b), suggest that the hub hexagons (ring a) and the six outer hexagons (e) hold a pronounced aromatic character, as shown by the blue shaded benzene rings, while the hetero pentagons (d) and the hexagons (c) are faintly aromatic. The hexagons (b) possess small positive average NICS(1)$_{zz}$ values, thus indicating a nearly non-aromatic character. Note that the hexagons (c) in *p*-HBC becomes non-aromatic, thus indicating the aromaticity distribution in each bowl system is nearly, yet not completely similar to that in *p*-HBC[55]. Such aromaticity distribution characteristics are also consistent with Clar sextet rule[56] because the largest number of aromatic sextets can be found in the resonance hybrid when aromatic sextets locate at rings (a and e) overwhelm rings (d and c) (Supplementary Figs. 44 and 45). ACID plots of **1a**' and **1b**' revealed that clockwise 6π-electron local current pathways are presented in the hub (a) and rim (e) benzene rings. On the contrary, no significant local currents were observed in rings (b, c and d) of neither bowl. Remarkably, a clockwise global current was observed along 30π-electron pathway consisting of the lone pair electrons on S or Se atoms, which are caused by the superposition of these local currents in each molecule (Fig. 5c, d). Taken together, these outcomes lend further support to the aromaticity distribution characteristics obtained by NICS calculations.

The electrostatic potential (ESP)[57,58] of **1a**' (Fig. 6a, b) and **1b**' (Fig. 6d, e) were also calculated at B3LYP/6-311G(d,p) level of theory. Unlike planar PAHs containing two identical faces, the ESP maps of these bowl-shaped compounds illustrate that the convex faces display more negative values compared with their concave faces, corresponding to the directions of the intrinsic dipole moment from the convex surface to the concave surface. The calculated dipole moments (M062x/6-31G(d)) are 3.59 Debye for **1a**' (Fig. 6c) and 3.45 Debye for **1b**' (Fig. 6f), much higher than those of coranlunene (2.19 Debye)[59] and surmarnene (2.7 Debye)[60], likely due to the enlarged π-surface and depth of these supersized bowls. In addition, the incorporation of thiophene or selenophene subunits enhances their concave-convex polarization effect.

Bowl-to-bowl inversion of two bowls was investigated by variable temperature $^1$H NMR measurements in 1,2-dichlorobenzene-$d_4$ from 30−180 °C (Supplementary Figs. 40 and 41) and no alteration on chemical shifts were observed, reflecting their inability to undergo bowl-inversion at the temperatures screened and thus their high inversion barriers. The DFT calculations for inversion barriers of the unsubstituted **1a**' and **1b**' at the M062x/6-31g(d) level of theory revealed that the inversion would proceed via a planar transition state like corannulene[61] and sumanene[62], with inversion barriers (Δ$G^‡$) of 70.2 kcal/mol and 51.0 kcal/mol for **1a**' and **1b**', respectively (Supplementary Fig. 42 and Supplementary Tables 8, 9 and 10). Such high energy barriers suggest that the bowl-to-bowl inversion process might be impossible at ordinary temperatures according to the Eyring equation (Supplementary Fig. 43). So, their bowl-shaped π-systems are conformationally locked at mild temperatures.

## Optical and electrochemical properties

To elucidate the photophysical properties of both sulfur and selenium isologous, their UV-vis absorption and fluorescence spectra were recorded in dichloromethane with hexa-*tert*-butylhexa-*peri*-hexabenzocoronene (ᵗBu-HBC) as a reference (Fig. 7a). Both absorption spectra of **1a** and **1b** displayed a similar pattern with the maximum absorbance at around 400 nm and a faint hump extending up 510 nm, which are bathochromic absorption shifts *ca.* 40 nm compared to ᵗBu-HBC. These observations implied that the displacement of sulfur by selenium only changes little about the positions of bands. The spectroscopic similarity of two isologous stems from their similar structures in terms of geometrical and electronic aspects, as revealed by

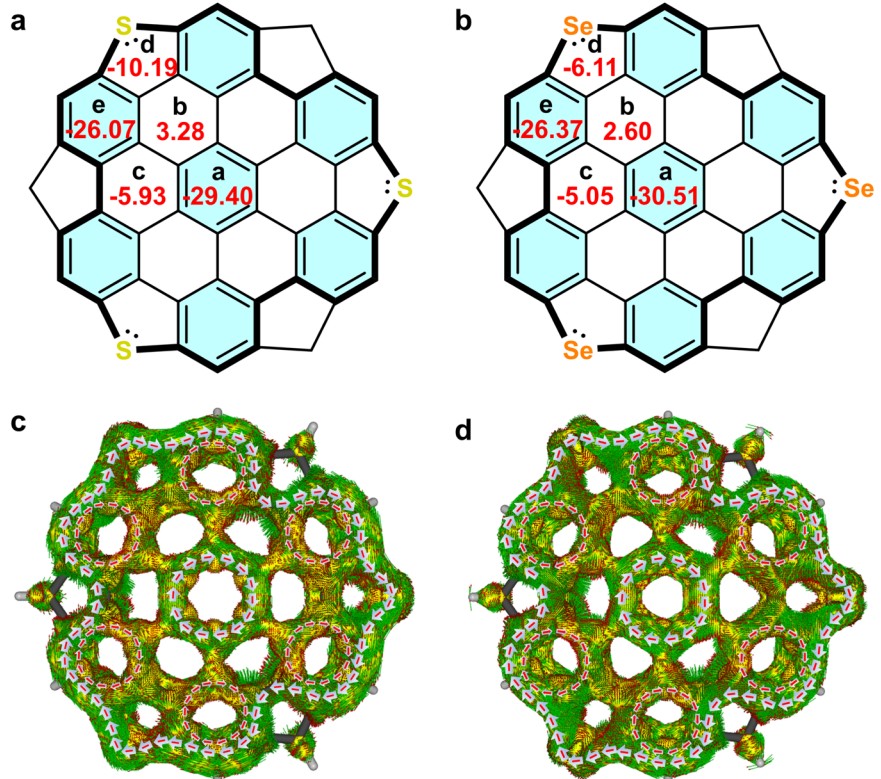

**Fig. 5 | NICS and ACID calculations of 1a′ and 1b′.** The average NICS(1)$_{zz}$ values (ppm, red) of **1a′** (**a**) and **1b′** (**b**). Clar sextets with pronounced aromatic characters are shaded in blue. Calculated ACID plots (isovalue = 0.03) of **1a′** (**c**) and **1b′** (**d**). Only contributions from π-electrons of the aromatic cores are considered. Red arrows indicate the directions of the induced ring current.

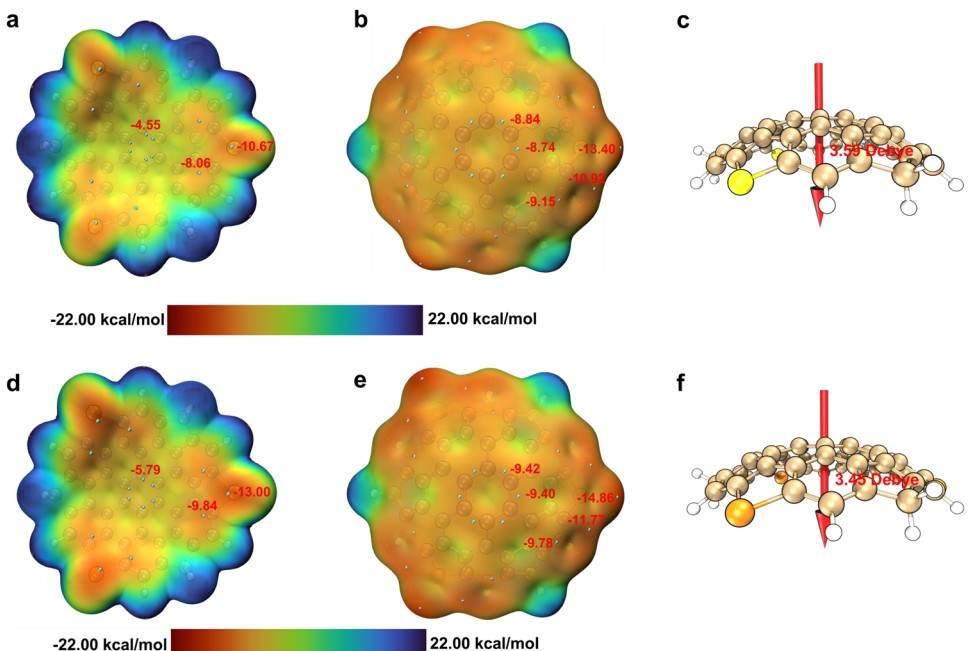

**Fig. 6 | Electrostatic potential and dipole moment calculations of 1a′ and 1b′.** Electrostatic potential maps of bowl concave for **1a′** (**a**) and **1b′** (**d**). Electrostatic potential maps of bowl convex for **1a′** (**b**) and **1b′** (**e**). Dipole moments of **1a′** (**c**) and **1b′** (**f**). Electrostatic potentials calculated on the 0.001 au. isodensity surface together with symmetry-averaged values (red) of local minima points (cyan). Red vector arrows indicate the directions of dipole moments from the negative to the positive charge.

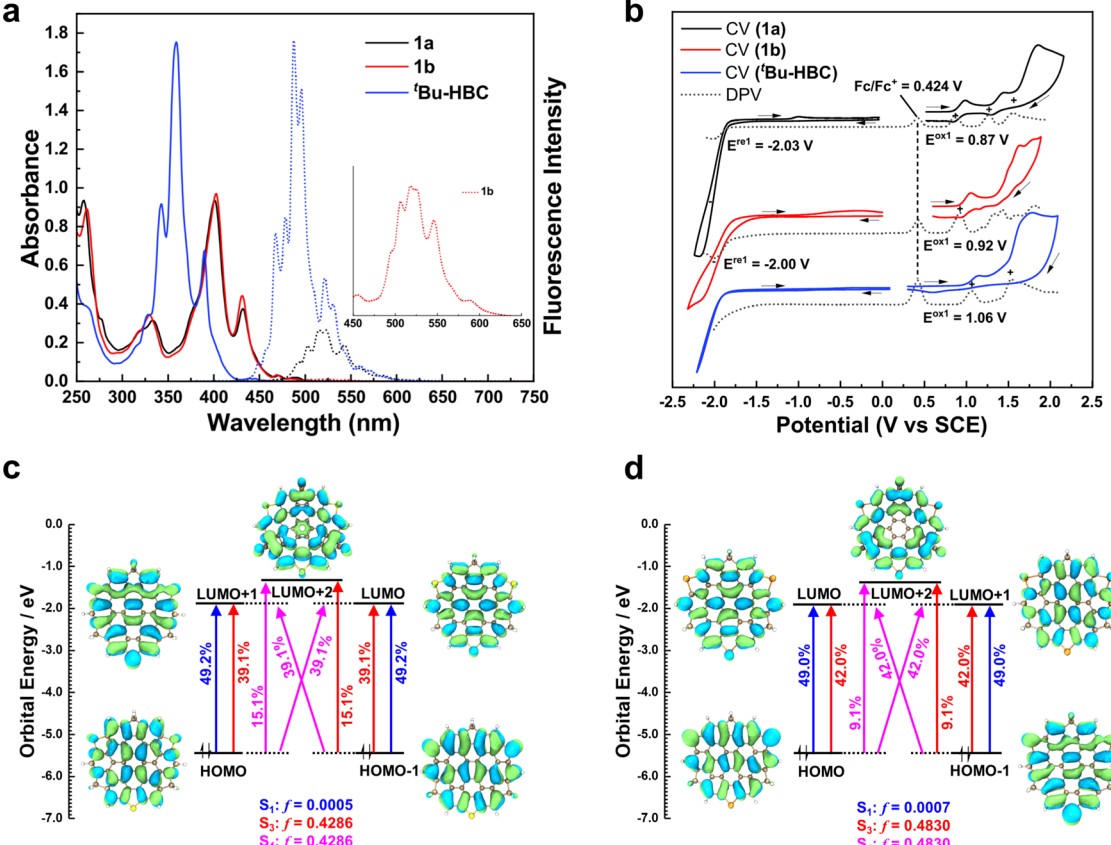

**Fig. 7 | Photophysical and electrochemical characterization and TD-DFT calculations. a** UV-Vis absorption (solid curves) and emission spectra (dotted curves) of **1a**, **1b** and ᵗBu-HBC in CH₂Cl₂ at a concentration of 1.0 × 10⁻⁵ M for absorption and 1.0 × 10⁻⁶ M for emission, respectively. **b** Cyclic voltammogram and differential pulse voltammogram of **1a**, **1b** and ᵗBu-HBC in CH₂Cl₂ (0.1 mol/L n-Bu₄NPF₆) at a scan rate of 0.1 V/s. All potentials were calibrated versus an aqueous SCE by the addition of ferrocene as an internal standard taking $E_{1/2}$ (Fc/Fc⁺) = 0.424 V vs. SCE[65]. Orbital correlation diagram and transition composition of the S₀ → (S₁, S₃, S₄) excited states for **1a'** (**c**) and **1b'** (**d**) calculated.

X-ray structural analysis and DFT calculations described above. TD-DFT calculations at the PBE/def2tzvp level for models **1a'** and **1b'** (Fig. 7c, d, Supplementary Tables 13 and 14) revealed that the maximum absorbance of **1a** is caused by the S₀ → S₃ and S₀ → S₄ transitions of **1a'**, which contain those from frontier MOs to nondegenerate LOMO+2 ($f = 0.4286$). While the lowest energy absorption band can be attributed to the S₀ → S₁ transition ($f = 0.0005$). The low intensity of S₀ → S₁ transition is related to the degeneracy of frontier molecular orbitals involved in the transition. The same rationale could be applied to explain the UV-vis spectrum of **1b** with similar molecular orbital energy level distribution in **1b'**. The optical energy gaps were estimated to be 2.43 eV (**1a**) and 2.45 eV (**1b**) from the onset wavelength of lowest energy absorption edge (Supplementary Table 3). The fluorescence spectra of both isologous featured distinctive vibronic structures with multiple maxima at 503, 515, 523, and 543 nm and bathochromic emission shifts *ca.* 30 nm compared to ᵗBu-HBC (Fig. 7a). The **1a** displayed a weak fluorescence while selenium isolog **1b** was found to be almost non-emissive due to the stronger heavy atom effect of selenium than sulfur. These results are explicable to that the degeneration from S₁ is symmetry forbidden, thus leading to the excited state relaxation via different vibronic levels of S₀. The absolute quantum yield and fluorescence lifetime of **1a** were measured to be 4% and 1.9 ns, the corresponding radiative ($k_f$) and nonradiative ($k_{nr}$) decay rate constants of **1a** were calculated from the singlet excited state to be 2.3 × 10⁷ s⁻¹ and 5.0 × 10⁸ s⁻¹, respectively. It is worth noting that comparable quantum yield but a significantly reduced lifetime of **1a** were observed in comparison with ᵗBu-HBC, which can be attributed to the low stability of exciting state due to curved π system (Supplementary Table 3).

Cyclic and differential pulse voltammetric experiments (Fig. 7b) were conducted to demonstrate the electrochemical properties of **1a** and **1b** with ᵗBu-HBC as a reference. The cyclic voltammetry of **1a** gave the first reversible and second quasi-reversible as well as third irreversible oxidation wave. In comparison, **1b** displayed the first quasi-reversible and subsequent multi-irreversible oxidation waves. The low reversibility of the oxidation waves for **1b** compared with **1a** may be ascribed to a stronger electron-rich characteristic of selenium atoms. The first half-wave potentials $E_{ox}^{1/2}$ of **1a** and **1b** locate at 0.87 V and 0.92 V against SCE, respectively, both of them are lower than ᵗBu-HBC ($E_{ox}^{1/2} = 1.06$ V) due to the heteroatoms doping. In addition, **1a** displayed negatively shifted first oxidation potentials compared to **1b**, which can be attributed to the better p-π conjugation and electron-donating effect of sulfur compared with selenium. All of the HOMO/LUMO energy levels were estimated from the onsets of oxidation from CV and approximate onsets of reduction from the DPV data, which displayed −5.23/−2.35 eV for **1a** and −5.30/−2.38 eV for **1b**, corresponding to the electrochemical energy gaps of 2.88 eV and 2.92 eV respectively (Supplementary Table 4), following their optical gaps and HOMO-LUMO gaps of DFT calculations (Supplementary Figs. 47 and 50).

## Host-guest chemistry with fullerenes

Due to the requirement for compatibility of Scholl reaction in the synthesis steps, the $sp^3$ bridge carbon must be assembled with geminal alkyl chains. Although the butyl groups endow **1a** along with precursor **4** sufficient solubility for purification and characterization, the steric hindrance of butyl groups prevents the intermolecular interaction between the curved surfaces of **1a** and fullerenes. However, the

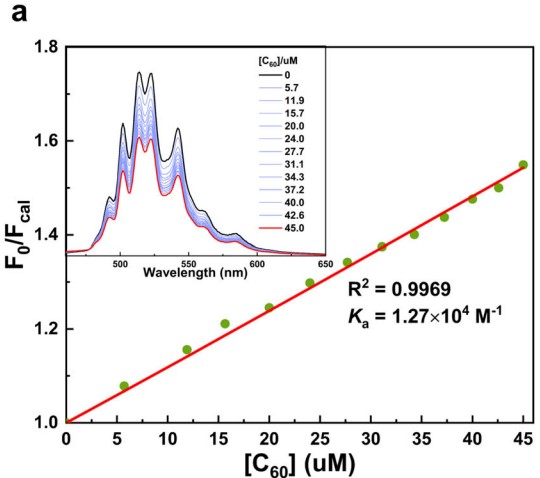

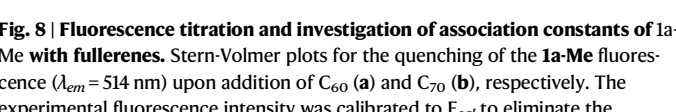

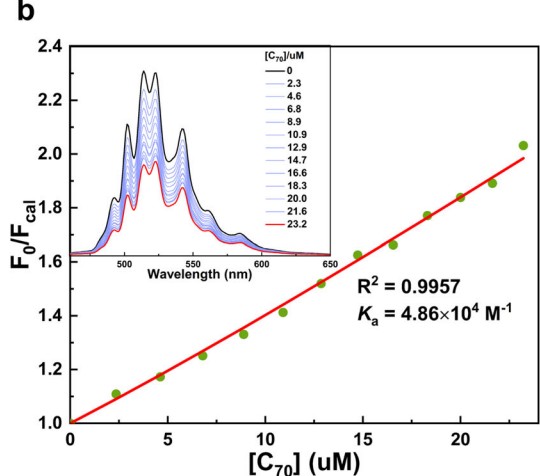

**Fig. 8 | Fluorescence titration and investigation of association constants of** 1a-**Me with fullerenes.** Stern-Volmer plots for the quenching of the **1a-Me** fluorescence ($\lambda_{em}$ = 514 nm) upon addition of $C_{60}$ (**a**) and $C_{70}$ (**b**), respectively. The experimental fluorescence intensity was calibrated to $F_{cal}$ to eliminate the competitive absorption of fullerene[66]. Association constant $K_a$ was calculated based on the nonlinear Stern-Volmer equation for 1:1 binding mode[63]. Insets: Fluorescence spectra of **1a-Me** upon titration with increasing amounts of fullerene guest, while the concentrations of **1a-Me** were kept constant at 6.0 uM.



**Fig. 9 | X-ray crystal structures of host-guest complexes.** The side views of host-guest complex **1a-Me@C$_{60}$** (**a**) and **1a-Me@C$_{70}$** (**b**, **c**). The shortest distance between fullerenes and hub rings of **1a-Me** (red). Solvents have been omitted for clarity. Thermal ellipsoids are shown at 30% probability.

electron-rich character of sulfur could change the inherent electronic properties of *p*-HBC framework and result in binding affinity toward electron-accepting fullerenes. To further investigate the association behavior of trithiasupersumanene with fullerenes, a methyl analog **1a-Me** was synthesized based on a similar protocol with **1a** and the association behavior with fullerenes was first monitored by ¹H NMR analysis. Upon the addition of $C_{60}$ or $C_{70}$ into a 1,2-dichlorobenzene-$d_4$ solution of **1a-Me**, the signals of aromatic protons and methyl groups inward bowl were slightly upfield and downfield shifted, respectively, and a considerable amount of black precipitate was formed in NMR tubes (Supplementary Fig. 34). These results suggest the formation of the host-guest complexes between **1a-Me** and fullerenes. Because the association constant could not be determined by NMR titration due to the low solubility of complexes, we used fluorescence titration for quantitative analysis. As illustrated in Fig. 8a, b, titration of **1a-Me** with $C_{60}$ or $C_{70}$ led to the gradual quenching of fluorescence intensity in 1,2-dichlorobenzene with negligible modifications of fluorescence lifetime (Supplementary Fig. 35). Therefore, the detected quenching behavior should be related to the formation of complexes (static quenching) rather than collisional quenching. At the same time, Job's plot experiment demonstrates a 1:1 host-guest stoichiometry for both $C_{60}$ and $C_{70}$ (Supplementary Figs. 32 and 33). The results were consistent with those of APCI-HRMS experiments, in which the peaks at *m/z* 1452.1053 and 1572.1061 corresponding to ([**1a-Me@C$_{60}$**]⁺, 1452.1035) and ([**1a-Me@C$_{70}$**]⁺, 1572.1035) respectively were found (Supplementary Figs. 83 and 84). Based on the nonlinear Stern-Volmer equation for 1:1 binding mode[63], the association constants were determined to be $K_a(C_{60}) = 1.27 \times 10^4$ M⁻¹ and $K_a(C_{70}) = 4.86 \times 10^4$ M⁻¹ by fitting the data

from the plots of $F_0/F_{cal}$ *vs*. the concentration of fullerene (Fig. 8a, b). It is noteworthy that a slightly stronger association constant for $C_{70}$ in comparison with $C_{60}$ was observed, which was further confirmed by gas-phase binding energy calculation. The binding energy for **1a-Me@C$_{70}$** complex (36.1 kcal/mol) is 1.6 kcal/mol higher than that of **1a-Me@C$_{60}$** (Supplementary Table 15).

Direct experimental evidence of the complexations was obtained by cocrystals suitable for X-ray diffraction analysis. As shown in Fig. 9, the crystal structures of **1a-Me@C$_{60}$** and **1a-Me@C$_{70}$** reveal that the formation of the complex with 1:1 stoichiometry, which are similar to those observed in solution. Notably, in contrast to the single $C_{60}$ conformation that exists in **1a-Me@C$_{60}$**, there are equal amounts of upright and latericumbent conformations for $C_{70}$ as enantiomer mixtures in a unit cell of **1a-Me@C$_{70}$** (Supplementary Fig. 29), which should be attributed to the lower symmetry of $C_{70}$ compared with $C_{60}$. The shortest distance between fullerene and the mean plane of the central six-membered ring of **1a-Me** is 3.12 Å for $C_{60}$ (Fig. 9a) and 3.16–3.25 Å for $C_{70}$ (Fig. 9b, c). All distances are shorter than the sum of van der Waals radii, thus indicating strong and multiple $\pi\cdots\pi$ interactions between **1a-Me** and fullerene. In addition, three alternating methyl groups of concave also interact with fullerenes through strong C−H $\cdots$ $\pi$ interactions (Supplementary Figs. 25 and 28). Apparently, it is the multiple non-covalent bonding that provides the driving force for the formation of complexes between **1a-Me** and fullerenes.

In conclusion, we have paved a concise 3-step approach for expeditious access to large, compact, and symmetric trichalcogen-asupersumanenes that underscore the following features: (1) a bowl-like architecture nanosized, 19 rings, 48 atoms, and 36 pyramidalized

trigonal carbon atoms; (2) a coronene core fully circumscribed by alternate hexagonal and pentagonal rings. Single-crystal X-ray crystallography and NMR spectroscopy clearly confirmed their bowl-shaped geometry in crystal and in solution. DFT calculations and variable temperature [1]H NMR experiments suggest their high inversion barriers. Quantum chemical calculations also reveal their aromaticity distribution and electrostatic potential characteristics. The large and rich-electron concave of trithiasupersumanene (**1a-Me**) render it an affinity to form 1:1 stable complexes with fullerenes $C_{60}$ or $C_{70}$ through $\pi \cdots \pi$ and multiple intramolecular C–H $\cdots \pi$ interactions. The success of the synthesis of these previously unknown superbowls delivered a new family of $\pi$-bowls and verified the accessibility of bridging all bays of *p*-HBCs. We believed that it will promote more synthetic efforts toward their all-carbon parent, other hetero versions as well as the analogs based on other PAHs with bay regions.

## Methods

### Synthesis

Selected procedures are shown below. Other procedures as well as compound data are described in the Supplementary Information.

1,3,5-tris(9,9-dibutyl-2,7-dichloro-9*H*-fluoren-4-yl)benzene (**3**). In a glove box, 1-(9,9-dibutyl-2,7-dichloro-9*H*-fluoren-4-yl)ethan-1-one **2** (390 mg, 1.0 mmol), $TiCl_4$ (0.7 mL, 6.6 mmol) and 5 mL dry *o*-dichlorobenzene were added to a 10 mL microwave vial and the vial was capped. The vessel was removed from the glove box and placed into a microwave reactor where it was heated at 180 °C for 3 h. After cooling down, the mixture was poured over concentrated hydrochloric acid/ice to quench the reaction and then extracted with $CH_2Cl_2$. The combined organic phase was washed with saturated aqueous $NaHCO_3$ and dried over $Na_2SO_4$. The organic solvent was removed at reduced pressure to give a yellow-brown solid. The above procedure was repeated 4 times and the crude material from each reaction was combined. The combined crude material was purified by column chromatography over silica gel (eluent: petroleum ether) to afford 1,3,5-tris(9,9-dibutyl-2,7-dichloro-9*H*-fluoren-4-yl)benzene **3** (970 mg, 52%) as an off-white powder; melting point (m.p.), 147–148 °C.

Trifluorenocoronene (**4**). To a mixture of **3** (220 mg, 0.2 mmol) and DDQ (360 mg, 1.6 mmol) in 1,2-dichloroethane (20 mL) was added trifluoromethanesulfonic acid (1 mL) under argon atmosphere, and the mixture was stirred at 50 °C for 2 h. After cooling to room temperature, the reaction mixture was quenched by adding saturated aqueous $NaHCO_3$, and then the mixture was extracted with $CH_2Cl_2$. The combined organic layer was dried over $Na_2SO_4$, and the solvent was removed under reduced pressure. The crude product was purified by silica gel column chromatography (eluent: petroleum ether) to give the trifluorenocoronene **4** (80 mg, 37%) as a yellow powder; m.p. > 300 °C.

Trithiasupersumanene (**1a**). In a glovebox, to an oven-dried pressure vessel with a Teflon screw cap was added compound **4** (110 mg, 0.1 mmol), $Bu_3SnSSnBu_3$ (245 mg, 0.4 mmol) and $Pd(PPh_3)_4$ (115 mg, 0.1 mmol) and 5 mL dry and degassed toluene. After the vessel was resealed and moved out from the glovebox, the mixture was heated and stirred at 150 °C for 48 h. On cooling to room temperature, the reaction mixture was quenched with saturated aqueous KF solution and extracted with $CH_2Cl_2$. The combined organic phase was dried over $Na_2SO_4$ before being filtered and concentrated down to a solid under reduced pressure. The crude solid was adsorbed onto silica gel and subjected to silica gel column chromatography (eluent, petroleum ether) to afford trithiasupersumanene **1a** (57 mg, 58% yield) as a yellowish powder; m.p. > 300 °C.

Triselenosupersumanene (**1b**). In a glovebox, to an oven-dried pressure vessel with a Teflon screw cap was added compound **4** (110 mg, 0.1 mmol), $Bu_3SnSeSnBu_3$[48] (263 mg, 0.4 mmol) and $Pd(PPh_3)_4$ (115 mg, 0.1 mmol) and 5 mL dry and degassed toluene. After the vessel was resealed and moved out from the glovebox, the mixture was heated and stirred at 150 °C for 48 h. On cooling to room

temperature, the reaction mixture was quenched with saturated aqueous KF solution and extracted with $CH_2Cl_2$. The combined organic phase was dried over $Na_2SO_4$ before being filtered and concentrated down to a solid under reduced pressure. The crude solid was adsorbed onto silica gel and subjected to silica gel column chromatography (eluent, petroleum ether) to afford triselenosupersumanene **1b** (44 mg, 39% yield) as a yellowish powder; m.p. > 300 °C.

Trithiasupersumanene **1a-Me** was obtained in a yield of 28% according to the similar protocol with **1a**. Full experiment details can be found in the Supplementary Information.

### Theoretical calculations

All calculations were performed using the *Gaussian 09, Revision D.01* program[64]. Geometric optimization, NICS[53], and ACID[54] calculations were conducted at the B3LYP/6-311G(d,p) level. Electrostatic potential calculations[57] were conducted at the B3LYP/6-311G(d,p) level and some calculated results were processed by Multiwfn[58]. Dipole moment and inversion barrier calculations were conducted at M062x/6-31G(d) level. TD-DFT calculations were conducted at PBE/def2tzvp level. Binding energy calculations were conducted at B3LYP-D3/6-311G(d,p) level.

## Data availability

X-ray crystallographic data (Supplementary Data 1) for compounds **1a**, **1b**, **1a-Me@C$_{60}$**, and **1a-Me@C$_{70}$** are freely available from the Cambridge Crystallographic Data Center via www.ccdc.cam.ac.uk/data_request/cif. (CCDC 2205194, 2205195, 2257288, and 2257286, respectively). All other data are available from the corresponding author upon request.

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

## Acknowledgements
J.W. thanks the National Science Foundation of China (NSFC) (Grant Nos. 21871169); J.L. thanks the National Science Foundation of China (NSFC) (Grant Nos. 21702131); Y.S. thanks the Fundamental Research Funds for the Central Universities (SNNU 2019TS040).

## Author contributions
Y.S., X.W., and B.Y. conducted the experiments with help from M.C., Y.W., Ji.L., M.X., and Y.Z. Y.S. performed the most DFT calculations. Z.G. and J.D. performed the inversion barrier calculations. H.S. collected and processed X-ray diffraction data. J.F. collected high-resolution mass spectrometry. J.W. conceived the original idea, designed and supervised the whole study. J.W., Y.S., and J.L. performed the data analysis and wrote the manuscript with feedback from others. All authors discussed the results and commented on the manuscript.

## Competing interests
The authors declare no competing interests.
