## [Peer Review File · Nature Communications]

Trichalcogenasupersumanenes and its concave-convex supramolecular assembly with fullerenesReviewers' Comments:

Reviewer #1:

Remarks to the Author:

This manuscript reports the synthesis of new large buckybowl molecules based on a hexabenzocoronene core. In general, giant buckybowls are very difficult to synthesize due to their instability caused by ring strain. However, the elegant synthetic strategy in this manuscript solved this problem to successfully make the superbuckybowl molecules. The synthesis itself is worth of reporting in a prestigious journal. The authors also investigated many properties of the obtained molecules, albeit very conventional ones. I recommend this manuscript to be published in Nature Communications. However, the following issues should be addressed and resolved prior to publication.

1) Title: "Trichalcogenohexaindenocoronene" is not appropriate because one indene structure is fused to adjacent indene rings. In this case, it cannot be called "hexaindeno". This should be changed.

2) Page 4, left column: "This is the first time that such a supersized buckybowl was achieved" is wrong. The carbon nanocone reported in J. Am. Chem. Soc. 2019, 141, 13008 is much larger because it has 26 fused pentagon/hexagons.

3) Page 5, right column: I could not understand the sentence, "As shown in Figs 4(c) and 4(f), there are 4 circles, totally 36 different pyramidalized trigonal carbon atoms existed in these coronene-based superbowls, except for 18 atoms at the peripheral vertexes."

What does "4 circles" mean? Why is the number of trigonal carbon atoms 36? If I am correct, this molecular core has forty two(42) sp²-carbon and three(3) sp³-carbon atoms. What does "18 atoms at the peripheral vertexes" mean?

4) The stability of the super buckybowl compounds should be mentioned. For example, are they stable in oxygen or water?

5) Reference [15-29] are too roughly organized. In my opinion, "buckybowls with only carbon and hydrogen atoms" and "heteroatoms-embedded buckybowls" should be mentioned separately.

6) Some important references are missing. The following papers on pi-extended buckybowls should be cited.

6-1) The Würthner group reported a carbon nanocone: J. Am. Chem. Soc. 2019, 141, 13008–13012.

6-2) The Ito group reported some pi-extended azacorannulenes: Angew. Chem. Int. Ed. 2018, 57, 9818; Angew. Chem. Int. Ed. 2022, 61, e202112638; Nat. Commun. 2022, 13, 1498.

6-3) The Hatakeyama group reported a B₂N₂-embedded corannulene: J. Am. Chem. Soc. 2018, 140, 13562.

7) The references cited are a bit old (mostly before 2020). As this field is advancing quickly, please update the references to cite recent papers as well.

8) There are many grammatical errors and expressions that do not make sense. English should be polished by a native professional proofreader.

Reviewer #2:

Remarks to the Author:

This is an interesting manuscript which describes the synthesis of two novel bowl-shaped poly-cyclic aromatics endowed with three chalcogen (S or Se) atoms and three substituted methylene groups at the bay regions of well-known hexa-peri-hexabenzocoronene. The synthetic approach is original, involving only three-steps, namely an aldol cyclotrimerization, a Scholl reaction, and a Stille

dehydrogenation. The resulting compounds have an interesting C_{3v} symmetry which has been confirmed by X-ray crystallography. Furthermore, a full spectroscopic characterization has been performed and also some relevant properties which have been underpinned by theoretical calculations, thus provided a nice description of the new nanographene.

The work has been competently carried out and the results are sound, based in a new controlled synthetic approach. Therefore, I feel that the manuscript meets the criteria of novelty and quality to be accepted for publication after addressing the following points:

- 1) The authors should mention the essays carried out in order to improve the Scholl reaction yield as mentioned. This could be of interest for the specialized community working on this topic.
- 2) In Figure 8 the axes cannot be visualized in (a) and (b). This makes difficult to follow the CV discussion. In this regard, the wave of the first oxidation potential seems to be reversible but, the second one, should be considered as quasireversible. Furthermore, the oxidation potential values for pristine HBC (or alkyl substituted) as a reference, would provide the real effect of the heteroatoms on the electrochemical properties.
- 3) The HOMO-LUMO gap for both compounds (1a and 1b) are basically the same. A small difference is, however, noted in the CV data. However, I do not see the explanation given by the authors to the slightly better oxidation potential value for the S compared to Se. Furthermore, the HOMO value is slightly higher for the Se compound, which means a better donor character.
- 4) Perhaps this review could be of interest for ref. 11-13: *Angew. Chem. Int. Ed.*, 2012, 51, 7094-7101.

Reviewer #3:

Remarks to the Author:

This manuscript by Li, Wei, and co-workers describes the synthesis and properties of trichalcogenohexaindenocoronenes. The titled compound represents a curved hexabenzocoronene (HBC) derivative having six bridging units at the bay area. Incorporation of these bridging units affords fused five-membered rings, resulting in a significantly curved structure. The synthesis of a similar compound was examined by a different group in 2015 by subjecting dodecachlorinated HBC to the nucleophilic aromatic substitution reaction with sulfide. This precedented synthesis afforded a tri-bridged compound instead of the desired bowl-shaped compound. The authors have overcome this issue in this manuscript by connecting three fluorene units by palladium-catalyzed C-S coupling reactions. The fact that the authors succeeded in synthesizing such a big and beautiful bowl-shaped hydrocarbon is worthy of praise. The authors explored the fundamental properties of 1a and 1b, including X-ray diffraction analysis, spectroscopic measurements, cyclic voltammetry, and DFT calculations. The discussions are solid, and this reviewer agrees for the most part.

However, this reviewer thinks that this manuscript lacks the novelty which deserves publication in *Nature Communications*. First, although the synthetic strategy shown in this manuscript is elegant, all the reactions have been developed previously. Especially the fact that a palladium-catalyzed C-S coupling reaction, which is a key step in the current strategy, is a powerful tool to create distortion has been demonstrated for the synthesis of distorted perylene bisimide derivatives (ref. 35). Second, the fundamental properties of trichalcogenohexaindenocoronenes including their structural distortion, photophysical properties, redox responses, and aromaticity are within expectations. In conclusion, this reviewer will support the acceptance if the authors describe other outstanding properties of the trichalcogenohexaindenocoronenes (i.e. unique reactivity, ferroelectricity in the solid state, and efficient fullerene-binding). The last two topics may require synthesizing other trichalcogenohexaindenocoronene derivatives with smaller alkyl chains.

Comments to authors

1. How about comparing the structural factors (bond lengths and bond alternation) of trichalcogenohexaindenocoronenes with those of HBC, which will give fruitful insight into the effect of distortion.
2. The authors note that hexagons (b) present a feeble antiaromatic character. However, this reviewer

thinks this argument is overstated because the NICS(1)zz values are close to zero.

3. On page 8, "Eying" should be "Eyring".

4. On page 12, "1,3,5-Tris" should be "1,3,5-tris".

5. For the ^{13}C NMR spectrum of 3, two signals are missing. Please comment.

6. On page S28 in SI, "Erying" should be "Eyring".

7. The ^1H NMR spectrum of 1b exhibits some undefined signals in the up-fielded region (i.e. triplet at 2.4 ppm and weak signals in the range of 1.3–0.6 ppm). Furthermore, the ^{13}C NMR signals of 1b are weak. These two points should be improved.

Point-by-point response

Reviewer comments:

Reviewer #1 (Remarks to the Author):

This manuscript reports the synthesis of new large bucky bowl molecules based on a hexabenzocoronene core. In general, giant buckybowls are very difficult to synthesize due to their instability caused by ring strain. However, the elegant synthetic strategy in this manuscript solved this problem to successfully make the superbuckybowl molecules. The synthesis itself is worth of reporting in a prestigious journal. The authors also investigated many properties of the obtained molecules, albeit very conventional ones. I recommend this manuscript to be published in Nature Communications. However, the following issues should be addressed and resolved prior to publication.

Response:

We express our sincere thanks to this reviewer for his/her extensive reading and we are so delighted to see these supportive comments.

1) Title: "Trichalcogenohexaindenocoronene" is not appropriate because one indene structure is fused to adjacent indene rings. In this case, it cannot be called "hexaindeno". This should be changed.

Response:

Thank you very much for your comments and suggestions. Because the IUPAC name for **1a** and **1b** are rather onerous, it is appropriate to introduce a more convenient trivial name. As it happens, the all-carbon version of **1a** and **1b** belongs to a family of large polycyclic aromatic hydrocarbons (PAHs) in which pentagonal and hexagonal rings alternately encircle a coronene (superbenzene) core, we prefer to call it "supersumanene" based on the similar structural characteristic and larger size compared with sumanene. In this way, **1a** and **1b** can be called trichalcogenasupersumanenes due to the heteroatoms doping. We have modified the title and added a description in the revised manuscript as follows.

Title: " Trichalcogenasupersumanenes and its concave-convex supramolecular assembly with fullerenes"

Such bowl molecules feature a coronene core successively circumscribed by alternate hexagonal and pentagonal rings, we prefer to call it "(hetero)supersumanene" as a convenient trivial name based on similar structural characteristics and larger size compared with sumanene.

2) Page 4, left column: "This is the first time that such a supersized bucky bowl was achieved" is wrong. The carbon nanocone reported in *J. Am. Chem. Soc.* 2019, 141, 13008 is much larger because it has 26 fused pentagon/hexagons.

Response:

Thank you very much for your comments and suggestions. We have deleted it in the revised manuscript.

3) Page 5, right column: I could not understand the sentence, "As shown in Figs 4(c) and 4(f), there are 4 circles, totally 36 different pyramidalized trigonal carbon atoms existed in these coronene-based superbowls, except for 18 atoms at the peripheral vertexes." What does "4 circles" mean? Why is the number of trigonal carbon atoms 36? If I am correct, this molecular core has forty two(42) sp²-carbon and three(3) sp³-carbon atoms. What does "18 atoms at the peripheral vertexes" mean?

Response:

Thank you very much for your comments and suggestions. We apologize for the lack of sufficient explanation. As shown in the following figure, "4 circles" represents the range of trigonal carbon atoms with POAV angles from the central hub ring to the periphery of bowl molecule. Because 12 (not 18) atoms at the peripheral vertexes including the trigonal carbon atoms of peripheral vertexes and 3 sp³-carbon atoms as well as heteroatoms have no POAV angles. There are errors and vagueness in previous expressions, we have corrected them in the revised manuscript as follows.

- 6 species of 36 pyramidalized carbon atoms (X ≠ Y)
- 4 species of 36 pyramidalized carbon atoms (X = Y)
- 12 atoms at the peripheral vertexes (X, Y, ●)

As shown in Figs 4(c) and 4(f), there are six categories in a total of 36 different pyramidalized trigonal carbon atoms that existed in these trichalcogenasupersumanenes, except for 6 negligible pyramidalized sp² carbon atoms at the peripheral vertexes.

4) *The stability of the super bucky bowl compounds should be mentioned. For example, are they stable in oxygen or water?*

Response:

Thank you very much for your comments and suggestions. We have modified it in the revised manuscript as follows.

Meanwhile, these trichalcogenasupersumanenes are stable enough and can be stored in solution and solid under air atmosphere for weeks without any change.

5) *Reference [15-29] are too roughly organized. In my opinion, "buckybowls with only carbon and hydrogen atoms" and "heteroatoms-embedded buckybowls" should be mentioned separately.*

Response:

Thank you very much for your comments and suggestions. We have modified it in the revised manuscript as follows.

Owing to the huge inner strain of bowl-shaped hydrocarbons makes their synthesis a major challenge, the number of buckybowls is still relatively rare compared with flat PAHs. Hitherto the most literature-known hydrocarbon buckybowls were related to C₆₀^[11–19] or C₇₀^[20–26] fullerene fragments and their π -extended derivatives^[27–30]. In addition, some heteroatoms-embedded buckybowls involving B^[31], N^[32–38], P^[39], S^[40], Se^[41–43], etc. have also been synthesized and sought as model compounds and partial structures for higher heterofullerenes.

New added references in the manuscript:

[14] Reisch, H. A., Bratcher, M. S. & Scott, L. T. Imposing curvature on a polyarene by intramolecular palladium-catalyzed arylation reactions: A simple synthesis of dibenzo[*a,g*]corannulene. *Org. Lett.* **2**, 1427–1430 (2000).

[17] Sygula, A., Marcinow, Z., Guzei, I. & Rabideau, P. W. The first crystal structure characterization of a semibuckminster fullerene, and a novel synthetic route. *Chem. Commun.* **24**, 2439–2440 (2000).

[18] Dickinson, C. F., Yang, J. K., Yap, G. P. A. & Tius, M. A. Modular synthesis of a semibuckminsterfullerene. *Org. Lett.* **24**, 5095–5098 (2022).

[21] Hishikawa, S. *et al.* Synthesis of a C₇₀ fragment bucky bowl C₂₈H₁₄ from a C₆₀ fragment sumanene. *Chem. Lett.* **46**, 1556–159 (2017).

[22] Nishimoto, M. *et al.* Synthesis of the C₇₀ fragment bucky bowl, homosumanene, and heterahomosumanenes via ring-expansion reactions from sumanene. *J. Org. Chem.* **87**, 2508–2519 (2022).

[24] Gao, G. P. *et al.* Rational functionalization of a C₇₀ bucky bowl to enable a C₇₀: bucky bowl cocrystal for organic semiconductor applications. *J. Am. Chem. Soc.* **142**, 2460–2470 (2020).

[26] Tanaka, Y., Fukui, N. & Shinokubo, H. as-Indaceno[3,2,1,8,7,6-ghijklm]terrylene as a near-infrared absorbing C₇₀-fragment. *Nat. Commun.* **11**, 3873 (2020).

[27] Shoyama, K. & Würthner, F. Synthesis of a Carbon nanocone by cascade annulation. *J. Am. Chem. Soc.* **141**, 13008–13012 (2019).

[28] Zhu, Z. Z. *et al.* Rational synthesis of an atomically precise carboncone under mild conditions. *Sci. Adv.* **5**, eaaw0982 (2019).

[29] Amaya, T., Nakata, T. & Hirao, T. Synthesis of highly strained π -bowls from sumanene. *J. Am. Chem. Soc.* **131**, 10810–10811 (2009).

[30] Amaya, T., Nakata, T. & Hirao, T. Construction of a hemifullerene skeleton: A regioselective intramolecular oxidative cyclization. *Angew. Chem. Int. Ed.* **54**, 5483–5487 (2015).

[31] Nakatsuka, S., Yasuda, N. & Hatakeyama, T. Four-step synthesis of B₂N₂-embedded corannulene. *J. Am. Chem. Soc.* **140**, 13562–13565 (2018).

[34] Tokimaru, Y. Ito, S. & Nozaki, K. A hybrid of corannulene and azacorannulene: synthesis of a highly curved nitrogen-containing bucky bowl. *Angew. Chem. Int. Ed.* **57**, 9818–9822 (2018).

[35] Li, Q. Q. *et al.* Diazapentabenzocorannulenium: a hydrophilic/biophilic cationic bucky bowl. *Angew. Chem. Int. Ed.* **61**, e202112638 (2022).

[36] Wang, W., Hanindita, F., Hamamoto, Y., Li, Y. & Ito, S. Fully conjugated azacorannulene dimer as large diaza[80]fullerene fragment. *Nat. Commun.* **13**, 1498 (2022).

[37] Krzeszewski, T., Dobrzycki, Ł., Sobolewski, A. L., Cyrański, M. K. & Gryko, D. T. Bowl-shaped pentagon and heptagon embedded nanographene containing a central pyrrolo[3,2-*b*]pyrrole core. *Angew. Chem. Int. Ed.* **60**, 14998–15005 (2021).

[39] Furukawa, S. *et al.* Triphosphasumanene trisulfide: high out-of-plane anisotropy and Janus-type π -surfaces. *J. Am. Chem. Soc.* **139**, 5787–5792 (2017).

[40] Imamura, K., Takimiya, K., Otsubo, T. & Aso, Y. Triphenyleno[1,12-*bcd*:4,5-*b'c'd'*:8,9-*b''c''d''*]trithiophene: the first bowl-shaped heteroaromatic. *Chem. Commun.* 1859–1860 (1999).

[43] Qiu, Z. L. *et al.* Synthesis and interlayer assembly of a graphenic bowl with peripheral selenium annulation. *J. Am. Chem. Soc.* **145**, 3289–3293 (2023).

6) Some important references are missing. The following papers on pi-extended buckybowls should be cited. 6-1) The Würthner group reported a carbon nanocone: *J. Am. Chem. Soc.* 2019, 141, 13008–13012. 6-2) The Ito group reported some pi-extended azacorannulenes: *Angew. Chem. Int. Ed.* 2018, 57, 9818; *Angew. Chem. Int. Ed.* 2022, 61, e202112638; *Nat. Commun.* 2022, 13, 1498. 6-3) The Hatakeyama group reported a B₂N₂-embedde corannulene: *J. Am. Chem. Soc.* 2018, 140, 13562.

Response:

Thank you very much for your comments and suggestions. We have added these important references in the revised manuscript.

7) The references cited are a bit old (mostly before 2020). As this field is advancing quickly, please update the references to cite recent papers as well.

Response:

Thank you very much for your comments and suggestions. We have updated the references [11-43] to cite recent papers as references.

8) There are many grammatical errors and expressions that do not make sense. English should be polished by a native professional proofreader.

Response:

Thank you very much for your comments and suggestions. Lastly, we revised the whole manuscript carefully to avoid language errors. In addition, we consulted a professional editing service and asked several colleagues who are native English speakers to check the English. We believe that the language is now acceptable for the review process.

Reviewer #2 (Remarks to the Author):

This is an interesting manuscript which describes the synthesis of two novel bowl-shaped poly-cyclic aromatics endowed with three chalcogen (S or Se) atoms and three substituted methylene groups at the bay regions of well-known hexa-peri-hexabenzocoronene. The synthetic approach is original, involving only three-steps, namely an aldol cyclotrimerization, a Scholl reaction, and a Stille dehydrogenation. The resulting compounds have an interesting C_{3v} symmetry which has been confirmed by X-ray crystallography. Furthermore, a full spectroscopic characterization has been performed and also some relevant properties which have been underpinned by theoretical calculations, thus provided a nice description of the new nanographene. The work has been competently carried out and the results are sound, based in a

new controlled synthetic approach. Therefore, I feel that the manuscript meets the criteria of novelty and quality to be accepted for publication after addressing the following points:

Response:

We express our sincere thanks to this reviewer for his/her extensive reading and we are so delighted to see these supportive comments.

1) The authors should mention the essays carried out in order to improve the Scholl reaction yield as mentioned. This could be of interest for the specialized community working on this topic.

Response:

We feel great thanks for your nice suggestions and made the following changes in the revised manuscript.

We also briefly explored the improvement of Scholl reaction of **3** including temperature, reaction time, and stoichiometry of acid and finally found relatively satisfactory conditions, as denoted in Fig. 2. It is noteworthy that the presence of chlorine atoms is also critical to the success of the Scholl reaction, replacing the chlorine atoms with the hydrogen atoms will lead to complex products. Moreover, the hexafluorinated and hexabrominated versions of key intermediate TFC **4** could not be obtained via oxidative cyclization.

2) In Figure 8 the axes cannot be visualized in (a) and (b). This makes difficult to follow the CV discussion. In this regard, the wave of the first oxidation potential seems to be reversible but, the second one, should be considered as quasireversible. Furthermore, the oxidation potential values for pristine HBC (or alkyl substituted) as a reference, would provide the real effect of the heteroatoms on the electrochemical properties.

Response:

Thanks for your nice suggestions. We have remeasured the cyclic voltammetry curve to display the reversibility of each oxidation wave. Furthermore, we have synthesized *hexa-tert-butyl* substituted HBC derivate (*t*Bu-HBC) as a reference and added some discussions in the revised manuscript as follows.

Cyclic and differential pulse voltammetric experiments (Fig. 8b) were conducted to demonstrate the electrochemical properties of **1a** and **1b** with *t*Bu-HBC as a reference. The cyclic voltammetry of **1a** gave the first reversible and second quasi-reversible as well as the third irreversible oxidation wave. In comparison, **1b** displayed the first quasi-reversible and subsequent multi-irreversible oxidation waves. The low reversibility of the oxidation waves for **1b** compared with **1a** may be ascribed to a stronger electron-rich characteristic of selenium atoms. The first half-wave potentials $E_{ox}^{1/2}$ of **1a** and **1b** locate at 0.87 V and 0.92 V against SCE respectively, both of them are lower than *t*Bu-HBC ($E_{ox}^{1/2}=1.06$ V) due to the heteroatoms doping. In addition, **1a** displayed negatively shifted first oxidation potentials compared to **1b**, which can be attributed to the better p- π conjugation and electron-donating effect of sulfur compared with selenium. All of the HOMO/LUMO energy levels were estimated from the onsets of oxidation from CV and approximate onsets of reduction from the DPV data, which displayed -5.23/-2.35 eV for **1a** and -5.30/-2.38 eV for **1b**, corresponding to the electrochemical energy gaps of 2.88 eV and 2.92 eV

respectively (Table S4), following their optical gaps and HOMO-LUMO gaps of DFT calculations.

3) The HOMO-LUMO gap for both compounds (**1a** and **1b**) are basically the same. A small difference is, however, noted in the CV data. However, I do not see the explanation given by the authors to the slightly better oxidation potential value for the S compared to Se. Furthermore, the HOMO value is slightly higher for the Se compound, which means a better donor character.

Response:

We feel great thanks for your nice suggestions, we have remeasured the cyclic voltammetry curve and corrected the discussion as follows.

In addition, **1a** displayed negatively shifted first oxidation potentials compared to **1b**, which can be attributed to the better p- π conjugation and electron-donating effect of sulfur compared with selenium. All of the HOMO/LUMO energy levels were estimated from the onsets of oxidation from CV and approximate onsets of reduction from the DPV data, which displayed -5.23/-2.35 eV for **1a** and -5.30/-2.38 eV for **1b**, corresponding to the electrochemical energy gaps of 2.88 eV and 2.92 eV respectively (Table S4), following their optical gaps and HOMO-LUMO gaps of DFT calculations.

4) Perhaps this review could be of interest for ref. 11-13: *Angew. Chem. Int. Ed.*, 2012, 51, 7094-7101.

Response:

Thank you very much for your comments and suggestions. We have added this important reference.

[9] Bunz, U. H. F., Menning, S. & Martín N. *para*-Connected cyclophenylenes and hemispherical polyarenes: building blocks for single-walled carbon nanotubes? *Angew. Chem. Int. Ed.* 51, 7094–7101 (2012)

Reviewer #3 (Remarks to the Author):

This manuscript by Li, Wei, and co-workers describes the synthesis and properties of trichalcogenohexaindenocoronenes. The titled compound represents a curved hexabenzocoronene (HBC) derivative having six bridging units at the bay area. Incorporation of these bridging units affords fused five-membered rings, resulting in a significantly curved structure. The synthesis of a similar compound was examined by a different group in 2015 by subjecting dodecachlorinated HBC to the nucleophilic aromatic substitution reaction with sulfide. This preceded synthesis afforded a tri-bridged compound instead of the desired bowl-shaped compound. The authors have overcome this issue in this manuscript by connecting three fluorene units by palladium-catalyzed C–S coupling reactions. The fact that the authors succeeded in synthesizing such a big and beautiful bowl-shaped hydrocarbon is worthy of praise. The authors explored the fundamental properties of 1a and 1b, including X-ray diffraction analysis, spectroscopic measurements, cyclic

voltammetry, and DFT calculations. The discussions are solid, and this reviewer agrees for the most part. However, this reviewer thinks that this manuscript lacks the novelty which deserves publication in *Nature Communications*. First, although the synthetic strategy shown in this manuscript is elegant, all the reactions have been developed previously. Especially the fact that a palladium-catalyzed C–S coupling reaction, which is a key step in the current strategy, is a powerful tool to create distortion has been demonstrated for the synthesis of distorted perylene bisimide derivatives (ref. 35). Second, the fundamental properties of trichalcogenohexaindenocoronenes including their structural distortion, photophysical properties, redox responses, and aromaticity are within expectations. In conclusion, this reviewer will support the acceptance if the authors describe other outstanding properties of the trichalcogenohexaindenocoronenes (i.e. unique reactivity, ferroelectricity in the solid state, and efficient fullerene-binding). The last two topics may require synthesizing other trichalcogenohexaindenocoronene derivatives with smaller alkyl chains.

Response:

Thank you very much for your comments and suggestions. We are sorry for the innovation of the synthesis strategy is not fully reflected in the manuscript. Here we want to emphasize the most notable merits more clearly. We must acknowledge that the reactions of three key steps make use of existing synthetic conditions (although with a little modification), but the work is by no means a simple stitching of existing methods. The window for success at each step is small, and finding the right conditions and breaking through the synthetic route is our painstaking effort.

Firstly, the aldol cyclotrimerization reaction to build a carbon skeleton is a great innovation, because the potential intramolecular cyclization of the acetyl group may fail the reaction. [1] This reaction was found to result in a reduced yield upon scaling up, perhaps influenced by the undesired intramolecular cyclization. Meanwhile, as described in the literature, "this method for assembling substituted benzene rings with 3-fold symmetry works well for some ketones, but fails for others, and the factors responsible for the success or failure of the reaction have never been firmly established." [2] So, the success of the aldol cyclotrimerization reaction for this substrate provides a unique case for this strategy.

[1] Cho, Y. J. & Lee, J. Y. Thermally stable aromatic amine derivative with symmetrically substituted double spirobifluorene core as a hole transport material for green phosphorescent organic light-emitting diodes. *Thin Solid Films* **522** 415–419 (2012).

[2] Ansems, R. B. M. & Scott, L. T. Circumtrindene: a geodesic dome of molecular dimensions. rational synthesis of 60% of C₆₀. *J. Am. Chem. Soc.* **122**, 2719–2724 (2000).

Secondly, the situation for the Scholl reaction of intermediate **TFB 3** is completely different from what has been reported in 1,3,5-tris(2-biphenyl)ylbenzene, [3] the contraction of

the five-membered ring elongates the distance to form the six-membered ring, meanwhile, the inactivation and steric hindrance of the chlorine atoms both make oxidation cyclization more difficult and make the success of the reaction less certain before the design route. The presence of chlorine atoms is also critical to the success of the Scholl reaction. Replacing the chlorine atoms with the hydrogen atoms will fail the Scholl reaction and lead to complex products. Moreover, the hexafluorinated and hexabrominated versions of key intermediates could not be obtained via oxidative cyclization. So, the success of this reaction is a matter of luck and miracle. Nowadays, the introduction of multi-chlorine atoms into the bay region of *p*-HBC was only realized by electrophilic substitution strategy,^[4] so the success of this reaction provides an appealing strategy towards the controlled edge chlorination of *p*-HBC. The success of key intermediate **TFC 4** also opens the chance for other heteroatom doping and further functionalization.

[3] Feng, X. L., Wu, J. S., Enkelmann, V. & Müllen, K. Hexa-peri-hexabenzocoronenes by efficient oxidative cyclodehydrogenation: the role of the oligophenylene precursors. *Org. Lett.* **8**, 1145–1148 (2006).

[4] Tan, Y.-Z. *et al.* Atomically precise edge chlorination of nanographenes and its application in graphene nanoribbons. *Nat. Commun.* **4**, 3646 (2013).

Thirdly, although the introduction of sulfur is based on the existing methods,^[5] However, in the case of di-sulfur annulated perylene bisimide derivatives, the aromatic core frameworks have almost perfectly flat structures with ignorable mean plane deviations (MPDs) of 0.018 Å, from the least-squares plane defined by peripheral 20 atoms of the aromatic core. Therefore, the structure strain is so small for the introduction of sulfur atoms. In striking contrast, the incorporation of sulfur into our curved system faces more difficulties due to the high strain involved, so this method is not guaranteed to work before the experiment. The incorporation of sulfur into the curved system, such as trithiasumanene, usually involves the initial formation of six-membered 1,2-dithiin rings followed by copper-mediated desulfurization,^[6]

so it is the first demonstration that this method can be applied to the synthesis of bowl molecules and introduce curvature.

mean plane deviations (MPDs) of 0.018 Å

X-ray crystallographic structure of S-heterocyclic annelated perylene bisimide

[5] Qian, H. L., Liu, C. M., Wang, Z. H. & Zhu, D. B. S-heterocyclic annelated perylene bisimide: synthesis and co-crystal with pyrene. *Chem. Commun.* 4587–4589 (2006).

[6] Li, X. X. et al. Non-pyrolytic, large-scale synthesis of trichalcogenasumanene: a two-step approach. *Angew. Chem. Int. Ed.* 53, 535–538 (2014).

Finally, thank you very much for your valuable suggestions to remind us to synthesize other trichalcogenasupersumanene derivatives with smaller alkyl chains for other outstanding properties. According to your nice suggestions, we have synthesized trithiasupersumanene derivative with methyl chains (**1a-Me**) and demonstrated moderate binding ability towards C₆₀ and C₇₀. The crystal structures are shown below, meanwhile, the host-guest chemistry with fullerenes section has been added in the manuscript and Supplementary Information respectively.

Fig. 10 X-ray crystal structures of host-guest complex 1a-Me@C₆₀ (a) and 1a-Me@C₇₀ (b, c).

1) How about comparing the structural factors (bond lengths and bond alternation) of trichalcogenohexaindenocoronenes with those of HBC, which will give fruitful insight into the effect of distortion.

Response:

We feel great thanks for your nice suggestions, we have added some discussions in the revised manuscript as follows.

The peripheral and hub benzene rings of **1a** and **1b** possess the smallest BLA values, which are similar to the parent HBC (Supplementary Fig. S19). In addition, both **1a** and **1b** display elongated bond lengths for the hub and periphery but shortened bond lengths for the rim of

coronene core in comparison with parent HBC (Supplementary Fig. S20) due to the high inner strain of their bowl geometries.

The comparison of BLA in **1a**, **1b** and *p*-HBC based on the result of X-ray crystal structure analysis (Figs S8, S17 and S19).

Figure S20. The comparison of average bond lengths (Å) in **1a**, **1b** and *p*-HBC based on the result of X-ray crystal structure analysis.

2) The authors note that hexagons (b) present a feeble antiaromatic character. However, this reviewer thinks this argument is overstated because the NICS(1)_{zz} values are close to zero.

Response:

We feel great thanks for your nice suggestions, we have corrected them in the revised manuscript.

The calculated NICS(1)_{zz} values, as denoted in the corresponding rings (Figs 5(a) and 5(b)), suggest that the hub hexagons (ring a) and the six outer (e) hold a pronounced aromatic character, as shown by the blue shaded benzene rings, while the hetero pentagons (d) and the hexagons (c) are faintly aromatic. The hexagons (b) possess small positive average NICS(1)_{zz} values, thus indicating a nearly non-aromatic character. Note that the hexagons (c) in *p*-HBC become non-aromatic, thus indicating the aromaticity distribution in each bowl system is nearly, yet not completely similar to that in *p*-HBC^[55].

3) On page 8, "Eying" should be "Eyring".

Response:

We thank this reviewer for his/her careful checking. We have corrected it in the revised manuscript.

4) On page 12, "1,3,5-Tris" should be "1,3,5-tris"

Response:

We thank this reviewer for his/her careful checking. We have corrected it in the revised manuscript.

5) For the ^{13}C NMR spectrum of **3**, two signals are missing. Please comment.

Response:

We thank this reviewer for his/her careful checking, we have recollected the ^{13}C NMR spectrum of **3** as follows.

Figure S66. ^{13}C NMR spectrum of **3** (151 MHz, CDCl_3 , 298K).

6) On page S28 in SI, "Erying" should be "Eyring".

Response:

We thank this reviewer for his/her careful checking. We have corrected it in the Supplementary Information.

7) The ^1H NMR spectrum of **1b** exhibits some undefined signals in the up-fielded region (i.e. triplet at 2.4 ppm and weak signals in the range of 1,3–0.6 ppm). Furthermore, the ^{13}C NMR signals of

1b are weak. These two points should be improved.

Response:

We thank this reviewer for his/her careful checking, we have purified carefully and recollected the ^1H NMR and ^{13}C NMR spectrum of **1b** as follows.

Figure S72. ^1H NMR spectrum of **1b** (600 MHz, CDCl_3 , 298K).

Figure S73. ^{13}C NMR spectrum of **1b** (151 MHz, CDCl_3 , 298K).

Reviewers' Comments:

Reviewer #2:

Remarks to the Author:

The authors have nicely addressed the previous suggestions/concerns of this reviewer and, therefore, I feel that the manuscript in its present form meets the criteria of novelty and quality to be accepted for publication in Nature Communications.

Reviewer #3:

Remarks to the Author:

The authors have provided compelling arguments in response to my inquiries. Additionally, they have satisfactorily addressed the concerns of the other reviewers who posed insightful questions, providing clear and concise answers.

Therefore, I am confident that the manuscript in its current form meets the necessary criteria to be accepted for publication in Nat. Commun. without further revisions.

Reviewer comments:

Reviewer #2 (Remarks to the Author):

The authors have nicely addressed the previous suggestions/concerns of this reviewer and, therefore, I feel that the manuscript in its present form meets the criteria of novelty and quality to be accepted for publication in Nature Communications.

Response:

We express our gratitude to this reviewer for his/her extensive reading and we are delighted to see the recommendation for acceptance.

Reviewer #3 (Remarks to the Author):

The authors have provided compelling arguments in response to my inquiries. Additionally, they have satisfactorily addressed the concerns of the other reviewers who posed insightful questions, providing clear and concise answers. Therefore, I am confident that the manuscript in its current form meets the necessary criteria to be accepted for publication in Nat. Commun. without further revisions.

Response:

We express our gratitude to this reviewer for his/her extensive reading and we are delighted to see the recommendation for acceptance.